# Green Consumer Profiling and Online Shopping of Imperfect Foods: Extending UTAUT with Web-Based Label Quality for Misshapen Organic Produce

**DOI:** 10.3390/foods13091401

**Published:** 2024-05-02

**Authors:** Rara Dwi Oktaviani, Phaninee Naruetharadhol, Siraphat Padthar, Chavis Ketkaew

**Affiliations:** 1International College, Khon Kaen University, 123 Mittraphap Road, Amphur Muang, Khon Kaen 40002, Thailand; rara.o@kkumail.com (R.D.O.); phaninee@kku.ac.th (P.N.); siraphatpt@kkumail.com (S.P.); 2Center for Sustainable Innovation and Society, Khon Kaen University, 123 Mittraphap Road, Amphur Muang, Khon Kaen 40002, Thailand

**Keywords:** green consumers, consumer segmentation, shades of green, extended UTAUT, online green purchase intention, imperfect produce, e-commerce

## Abstract

Misshapen organic vegetables in the food supply chain can easily be discarded in the market if they do not meet aesthetic standards. E-commerce technology enables the distribution of imperfect organic produce from farmers to potential customers, mitigating agri-food waste issues. Green consumers are prospective customers of imperfect produce because their purchasing decisions are made based on sustainability and environmental considerations. However, each individual’s degree of green consumption differs, impacting their preferences and behaviors toward green buying activity. Considering the gap between the varying levels of green consumers and their intention to purchase imperfect organic produce through e-commerce, this study aimed to profile three distinct green consumers and explore factors affecting their Online Green Purchase Intention (OGPI) for imperfect organic vegetables. The Unified Theory of Acceptance and Use of Technology (UTAUT) was applied in this study, and Web-Based Label Quality (WLQ) was introduced as an extended construct to describe green consumers’ perceptions of the credibility and reliability of labels or product-related information displayed on e-commerce platforms. This study involved 668 internet users from environmental platforms and online communities of organic food enthusiasts. First, the consumers were classified into dark-green, semi/light green, and non-green using a cluster analysis approach. Then, Structural Equation Modeling (SEM) and Multi-Group Analysis (MGA) were employed to determine the factors affecting OGPI among green consumer groups. This research found that Performance Expectancy (PE), Social Influence (SI), and WLQ positively influenced dark-green consumers’ online green purchase intention. Only Performance Expectancy (PE) positively affected semi/light-green consumers’ OGPI. Meanwhile, the Facilitating Condition (FC) positively affected non-green consumers’ online green purchase intentions. This research revealed dark-green consumers as the target segment, broadening customers’ perspectives on accepting imperfect organic products, including e-commerce technology. Moreover, the research results can be utilized for marketing and business purposes and contribute to food policy.

## 1. Introduction

Globally, fruits and vegetables are the largest producers of waste among the food categories. Approximately 65 kg of food is wasted per year by one person, and the most contributed waste comprises 25% vegetables, 24% cereals, and 1% fruits [1]. Imperfect produce is one of the reasons why vegetables contribute to Fruit and Vegetable Waste (FVW) globally [2]. Naruetharadhol et al. (2023) mentioned that consumers hold a pessimistic outlook toward vegetables that possess imperfect shapes [3]. They avoid accepting unappealing vegetables due to the association between consuming such products and a negative perception of their own attractiveness, morality, and health [4,5]. Another prior study revealed that individuals anticipated unsatisfactory produce to have lower taste and nutritional value compared to satisfactory produce [3,6]. In organic produce markets, consumers’ negative perceptions toward anesthetic vegetables lead to an increasing demand for perfect-looking organic vegetables, explicitly contributing to the higher number of FVW from discarded imperfect organic produce. In fact, organic fruits and vegetables may exhibit visual imperfections due to the absence of pesticides or synthetic fertilizers in organic farming, which are typically used to enhance the aesthetic appeal of the produce [7]. However, the imperfection does not affect the taste or nutritional content of organic produce [4].

Additionally, consuming imperfect organic produce contributes to environmental sustainability by reducing food waste [8]. Individuals who are pro-environment or green consumers are more inclined to hold positive attitudes toward green purchases [9,10]. Green purchasing refers to the act of environmentally conscious consumers buying products that are green in order to save resources and protect the environment [11]. Nevertheless, consumers who prioritize green choices demonstrate varying consumption patterns, leading to various attitudes toward green purchases [12,13]. Susanty et al. (2022) examined the behaviors of three distinct green consumer groups (dark-green, green, and light-green) and the possibility of a shift in the degree of greenness toward green consumption [13]. It has been verified that every green consumer possesses distinct perspectives regarding environmental standards, and dark-green consumers exhibit the highest level of views compared to other groups. The level of pro-environmental behavior may influence people’s willingness to purchase imperfect organic vegetables.

In response to the problem of imperfect organic fruits and vegetables, which continue to contribute to waste due to their unaesthetic shape, e-commerce platforms can be used to tackle this problem by distributing them to prospective customers. Studying related factors influencing online green purchase intention toward imperfect products via e-commerce remains essential. Web-Based Label Quality (WLQ) is a newly introduced construct in this research that is considered more effective and captivating when encouraging people to buy imperfect organic vegetables to enhance food sustainability. The construct focuses on how consumers perceive the quality of labels or information displayed on e-commerce platforms, which encompasses accuracy, clarity, and completeness derived from the combination of platform credibility, trustworthiness, and online transactional processes [14,15,16,17]. WLQ is a novelty because it has yet to be thoroughly examined and mentioned in prior studies related to online green purchase intention [18,19].

The primary objective of this study is to investigate the behaviors and factors influencing green consumers’ intentions to purchase imperfect produce via e-commerce platforms. Specifically, it focuses on the behaviors of environmentally conscious consumers at varying levels of environmental awareness and examines the factors that affect their willingness to purchase misshapen organic fruits and vegetables online. The research employs an extended model of the Unified Theory of Acceptance and Use of Technology (UTAUT), which includes Performance Expectancy (PE), Effort Expectancy (EE), Facilitating Condition (FC), Social Influence (SI), and Web-Based Label Quality (WLQ) as an extended construct, with Online Green Purchase Intention (OGPI) as the dependent variable. Moreover, the study incorporates green consumer segmentation, as Polonsky (1995) proposed, using cluster analysis to categorize consumers into three segments: dark-green, semi/light-green, and non-green [20].

This study introduces Structural Equation Modeling (SEM) and Multi-Group Analysis (MGA) to predict purchasing habits across green consumer segments. Including three distinct consumer groups (dark-green, semi/light-green, and non-green) in the SEM analysis serves as a moderating effect, offering new insights into consumers’ intentions to buy imperfect produce from e-commerce platforms. The MGA using three consumer groups extends previous research typically conducted with only one or two green consumer groups [21,22,23]. The practical applications of these findings are significant for various stakeholders, including marketers, researchers, marketplace developers, and policymakers. By understanding segmented consumer behaviors and preferences, these beneficiaries can refine marketing and service strategies, develop more targeted digital markets, and contribute to the formulation of sustainable food policies. The case study of the Ugly Veggies platform in Thailand provides a contextual foundation for applying these insights in a real-world setting, enhancing the relevance and applicability of the research outcomes.

## 2. Literature Review and Related Works

The emergence of digital and online buying significantly influences food sustainability by increasing accessibility to green foods [24,25]. In the context of FVW, digital technology is a prospective solution to combat agri-food waste [26]. Digital technology mitigates agricultural food waste by facilitating the connection between sellers of imperfect produce and green consumers through e-commerce. This scenario enables the transactional processing of imperfect organic vegetables that would have otherwise been discarded due to their non-standard shape. Naruetharadhol et al. (2023) studied an e-commerce platform for combating food waste [3]. They suggested a new circular economy-based e-commerce platform for selling imperfect organic fruits and vegetables and explored the factors influencing users’ intentions to adopt an e-commerce platform. A prior study also expressed apprehension regarding the platform’s characteristics due to its potential influence on customers’ decisions to buy organic items through online channels [15]. The study highlighted that platform credibility has a more significant impact on perceived value, which means consumers are more concerned about the platform’s characteristics when shopping for organic food online. Additionally, Qalati et al. (2021) confirmed that service quality, website quality, and platform reputation generate online buying trust as the most powerful elements in online purchase intention [27]. Aside from making good use of IT, other aspects like website design and user interaction are crucial to e-commerce success [16]. The visual appeal of an online store’s website is a tool that helps communicate effectively with customers. Aesthetics is the most influential variable with respect to “visit,” “purchase,” and “comparison to similar products on other websites,” in that order. Thus, it is essential to consider the development of a digital platform that integrates elements of clear, accurate, and useful information display to enhance the user experience. E-commerce and delivery platforms focusing on imperfect food in the Western world have been the subjects of recent academic case studies and empirical research [28,29]. Nonetheless, this concept remains underexplored in emerging economies. Thailand has embraced the transition toward sustainability and has developed an e-commerce platform named Ugly Veggies Thailand, which is utilized as an empirical case in this study.

### 2.1. Ugly Vegetables e-Commerce Site

The Ugly Veggies Platform is an electronic commerce platform that facilitates the sale and purchase of imperfect organic fruits and vegetables from certified organic farms in Thailand, which has become a point of connection between farmers and consumers (see Figure 1). This platform has adopted the principles of the circular economy, which emphasize sustainability and promote resource regeneration. It particularly focuses on rescuing misshapen organic vegetables and fruits that would typically be discarded when they cannot be sold or supplied to customers. By establishing a connection, it becomes possible to efficiently distribute imperfect organic fruits and vegetables that are rich in nutrients and safe for consumption. This connection also helps optimize the supply chain by reducing the number of intermediaries involved in the process of getting produce from farmers to the market. This platform aims to fight the problem of food waste and the presence of large intermediaries in Thailand.

According to Figure 1, buyers have the option to purchase organic, unattractive vegetable items from three different integrated sources: the Ugly Veggies Web App, Line stores, and Facebook. The platform will have various certified farm shops, and customers can easily order the desirable organic products from their selected shop. Afterward, the administrator of Ugly Veggies will review and authorize the orders. When the admin accepts the request, the farmer will be notified to deliver the package quickly via the provided delivery service. The order will be received within a timeframe of 1–2 days.

### 2.2. Green Consumers and Segmentation

Green consumers are people who are aware of protecting the environment by selectively buying environmentally friendly products or services, avoiding goods that endanger the sustainability of the earth and the future of humankind, and maintaining their health and lifestyle [30]. Pro-environmental awareness is a significant base for green consumption [31]. Research found that people are willing to buy environmentally friendly products and services and are willing to spend extra money to make businesses more ecologically friendly [20]. However, green consumers have different attitudes when they are interested in buying environmental products. Some segmentation concepts are compared regarding their green behaviors.

Table 1 presents the classification of green customers by Organization and Wax (1990), who categorized them according to five levels of environmental concern and their purchasing activity [32]. Among these groups, Basic Brown exhibited the highest level of adherence. Meanwhile, Ottman (2010) categorized them into four distinct categories that were clustered according to their spirits: hate waste, health enthusiasts, animal lovers, and outdoor enthusiasts [33]. Nevertheless, this study will utilize the segmentation proposed by Polonsky (1995) because their characteristics and behaviors are closely related to green purchasing activity, which is in line with the purpose of understanding the customer’s online purchase intention of imperfect organic produce [20,30,34]. Dark-green refers to customers who actively seek green information, which influences their purchasing behavior. Their shopping lists had been meticulously planned [35]. Conversely, semi/light-green individuals occasionally prefer to consume environmentally friendly products, whereas non-green customers do not prioritize purchasing and using such products. Additionally, grouping three distinct groups focusing on broad segments offers greater clarity, actionability, and efficiency, especially in research that rarely conducts multigroup analysis on online green purchase intention.

### 2.3. Online Green Purchase Intention (OGPI)

The influence of online media and technological progress evolved the common buying intention into an online green purchase intention driven by the internal motive of environmental consciousness [22]. Green purchase intention is primarily concerned with purchasing eco-friendly items, ignoring those that harm the environment [36]. E-commerce platforms are crucial for promoting environmentally friendly purchasing by offering substantial information, enabling easy access to green items, and establishing a digital setting that prioritizes sustainability [37]. The convenience, accessibility, and wider range of products provided by digital and online shopping may enhance the probability of online green purchasing imperfect organic vegetables that initially contribute to food waste [10,26,38].

According to Štofejová et al. (2023), e-commerce has a significant impact on individuals’ willingness to pay for environmentally friendly products, influencing their environmental purchasing behavior and reinforcing their intentions to make future purchases [39]. Through the process of digitalization, businesses have the opportunity to encourage consumers to uphold their environmental beliefs and motivate them to adopt sustainable behaviors by offering accessibility for purchasing green items through online platforms. Teresa Foti et al. (2022) examined how sustainable consumption drivers affect agricultural product online buying intentions [40]. Improving consumers’ perceived values is crucial to sustainable consumption and online agriculture product purchases. To increase online purchase intention, online businesses must target specific customer groups, understand consumer psychology, and provide distinctive products in a specified order.

In the digital age, an online platform is considered an option for accessing eco-friendly products. Consumers search for products that employ sustainable methods and demonstrate environmental awareness [41]. In this study context, OGPI refers to consumers’ online buying intentions toward imperfect organic vegetables through e-commerce.

## 3. Theoretical Framework and Hypothesis Development

The following section will explain the factors influencing OGPI explored in this research. Several hypotheses will be employed based on the UTAUT theory and prior studies. Moreover, WLQ (a novel construct) will be defined to extend the UTAUT model and form the research framework.

### 3.1. Unified Theory of Acceptance and Use of Technology (UTAUT)

Technology acceptance theory has been massively developed with different sets of acceptance constructs. Venkatesh et al. (2003) developed User Acceptance of Information Technology toward a unified view called UTAUT theory [42]. This theory is derived from different disciplines, such as the Theory of Reasoned Action, Technology Acceptance Model, Motivational Model, Theory of Planned Behavior, Diffusion of Innovation, and Social Cognitive, embracing several variables reflecting different viewpoints and disciplines and expanding the applications of the theory to different contexts.. The UTAUT demonstrates superiority in comprehending the purpose of utilizing certain technology: it accounts for 70% of the acceptance of technology, whereas earlier models could only account for approximately 40%. It was formulated by four primary constructs: Performance Expectancy, Effort Expectancy, Social Influence, and Facilitating Conditions.

Several recent studies utilized the UTAUT theory to understand customers’ intentional usage and purchase intention online [1,43,44,45,46]. Erjavec and Manfreda (2022) applied the UTAUT theory to online shopping adoption during the pandemic and social isolation with an extended UTAUT construct, herd behavior consisting of imitating others and discounting one’s own information [43]. This study found that the e-commerce platform had significant potential in fresh food retail due to the lockdowns. Meanwhile, Chen et al. (2021) assessed customers’ online purchase intentions on the fresh e-commerce platform utilizing the UTAUT model [1]. They extended one construct, perceived risk, and implemented perceived trust as a mediator for each construct. Thus, food safety awareness was expected to mediate between trust and purchase intention. This research discovered that COVID-19 disrupts people’s daily lives and possibly changes the existing theoretical model. Therefore, UTAUT has become a qualified theory widely used in growing research to comprehend adoption intentions and purchase intentions toward technology based on customer behavioral tendencies.

This study expands the original UTAUT model with an additional construct, WLQ, by utilizing three moderating effects (dark-green, semi/light-green, and non-green). Then, we explore the different behaviors among three green consumers and their OGPI toward e-commerce selling imperfect produce.

#### 3.1.1. Performance Expectancy (PE)

Venkatesh et al. revealed that Performance Expectancy (PE) is involved in high performances [42]. PE made people believe that using the system would aid them in accomplishing goals related to individual performance.

Chen et al. (2021) studied factors influencing consumers’ intentions to purchase fresh food online with the UTAUT theory [1]. Fresh e-commerce sells various fresh foods online; most of them are perishable foods. The research revealed that PE had a positive influence on the fresh e-commerce platform. The consumers’ purchase intention is influenced by their expectations regarding the performance of the new e-commerce platform. The improved efficiency resulting from the fresh e-commerce platform significantly increased consumer preference for buying fresh products from the platform.

Accordingly, the customer’s expectation of e-commerce can impact their higher intention to make a purchase [47]. Enhanced efficiency may improve the customer’s inclination to purchase imperfect organic fruits and vegetables via e-commerce. Time savings, quick-finding items, and efficient grocery management are some factors that make customers prefer to buy organic food online [48]. Digital markets can fully support them daily. Those benefits can influence customers to use an e-commerce platform [1,3]. Thus, the researcher presents the following hypothesis:

**Hypothesis** **1** **(H1).**
*PE positively influences OGPI.*


#### 3.1.2. Effort Expectancy (EE)

Effort Expectancy (EE) is related to the ease of the system. The theory was captured from unified theories such as perceived ease of use (TAM/TAM2), complexity (MPCU), and ease of use (IDT) [42]. EE is responsible for creating a user-friendly interface and optimizing the customer experience. It is supposed to catch the customer’s initial impression during the online transaction in order to enhance their intention to make purchases through the platform [49].

Several research papers have shown that EE influences online purchase intention. Hong et al. (2023) studied the factors influencing Online Food Delivery Services (OFDS), which revealed that EE has no significant impact on online purchase intention [23]. The potential rationale for this result is that the convenience of the service no longer appeals to consumers due to the prevalent presence of online food delivery services on their mobile phones. It made them familiar with online service, resulting in nothing particularly noteworthy. On the other hand, a study conducted by an e-commerce platform selling fresh food hypothesized that EE has a beneficial impact on buying intention [1]. They believed that the platform had easy functionality and that it resulted in decreased learning expenses. Nevertheless, the findings indicated that the impact of EE was not significant. This is likely due to the fact that consumers who have become accustomed to the convenience and rapidity of online shopping in recent years are less responsive to minor technological improvements in a new e-commerce platform.

In the present context, EE pertains to the level of simplicity and convenience experienced by consumers while making an online green purchase on a new e-commerce platform. They can effortlessly operate it and comprehend the diverse features on the page. These designs encompass the ease of use for consumers to order imperfect organic fruits and vegetables from organic farmer stores on the platform and the promptness with which they receive resolutions when encountering specific issues on the platform. The ease of accessing the platform function positively affects views about green products, resulting in a rise in behavioral intention to purchase green items online [50]. Hence, the researcher proposes that the following:

**Hypothesis** **2** **(H2).**
*EE positively influences OGPI.*


#### 3.1.3. Facilitating Condition (FC)

Facilitating Condition (FC) describes user perceptions of assessing technology regarding the availability of organizational support and existing infrastructure [42]. The components included knowledge, resources, technology, and equipment [3,51]. In terms of e-commerce, FC refers to the degree to which customers have the necessary resources to engage in online shopping. For instance, the availability of internet mobile access and reliable platforms [42].

The presence of comprehensive infrastructure and technical components can lessen the gap between green purchase intention and green purchase behavior [52]. The condition occurs due to the convenient accessibility and comfort of the facilities offered to encourage individuals to adopt environmentally conscious consumption habits. Another study found that the firm’s e-service quality is strongly related to green buying intentions [53]. Understanding the client’s demands and offering them quick service, trustworthy, and tailored assistance will naturally be connected to establishing the consumer’s green buying intention.

Correspondingly, comprehensive facilities to support the operation of e-commerce are necessities. Users considered the technology useful and beneficial if they owned direct access to the relevant “infrastructures” [54]. Indeed, infrastructure, technical support, and knowledge from the consumer and firm sides may increase OGPI via e-commerce. As a result, the researcher proposes the following:

**Hypothesis** **3** **(H3).**
*FC positively influences OGPI.*


#### 3.1.4. Social Influence (SI)

The term “Social Influence (SI)” refers to an individual’s perception that influential individuals believe they ought to use the new system [42]. According to Pienwisetkaew et al. (2023), family, partners, and close friends might influence the expression, behavior, or adoption of technology [21]. A previous study presented that an individual’s social environment might have an impact on consumers’ dietary preferences [55] Evidently, Santaliestra-Pasías et al. (2022) indicated that adolescents who eat meals with their families tend to consume a greater number of nutritious foods and drinks [56]. It emphasizes that the dietary decisions of individuals are influenced by their closest associates.

Additionally, SI is one of the key determinant factors that influence customers’ OGPI [48]. Naruetharadhol et al. (2023) revealed that SI has a beneficial impact on a tendency to embrace an e-commerce platform, selling imperfect organic fruits and vegetables to health-conscious consumer groups [3]. The willingness to pay for unsatisfactory-shaped organic items becomes greater through the influence of friends or social media influencers. The credibility of influencers and the level of para-social interaction have a substantial impact on customers’ purchase decisions [57]. As mentioned, the finding is strengthened by the research from Chen et al. (2021), in which SI significantly impacted online purchase intention when family members or close friends encouraged them to buy fresh foods through the e-commerce platform [1]. These studies have demonstrated that individuals in one’s social circle have a significant impact on one’s intention to make online purchases.

Prior studies suggest that SI may have an impact on customers consuming organic food, which in turn affects customers’ OGPI through e-commerce. Therefore, the researcher suggests the following:

**Hypothesis** **4** **(H4).**
*SI positively influences OGPI.*


### 3.2. Extended UTAUT Theory

#### Web-Based Label Quality Perception (WLQ)

The development of digital technology has influenced consumer behavior research, particularly in the e-commerce area. A novel construct known as Web-Based Label Quality (WQL) has emerged to investigate the customer’s perspective related to the label of digital platforms in their online shopping experiences.

Web-Based Label Perception Quality (WLQ) encompasses several aspects of how consumers assess the credibility and reliability of labels or information pertaining to products shown on e-commerce platforms. The factors encompass graphical representation, data accuracy, reliability, comprehensibility, and influence over buying decisions. This concept perceives the integration of digital visuals, textual information, and psychological aspects that impact customer perceptions and actions in online markets.

According to food S-commerce research, credibility is the judgment of customers based on their simple evaluation of the platform [58]. The aesthetical platform significantly affects consumers’ decisions as well as their feelings [16]. It was mentioned that in order to attract and keep clients, online store operators should prioritize aesthetics when developing their websites. The results show how stimuli aesthetics have complicated effects and how important design aesthetics are in influencing consumers’ psychological and behavioral reactions. Additionally, the platform’s information quality and technical stability may boost platform credibility among users [58]. Because of the higher level of credibility, consumers have less fear about their privacy, which builds confidence and increases their online purchase intentions. In other words, a basic assessment of a certain S-commerce platform can boost corporate standards and empower trust, resulting in huge amounts of transactions.

WLQ, in the context of this study, is a novel concept that applies how consumers pertain to examining consumers’ perceptions of the quality of label information displayed on websites and e-commerce platforms. This construct is a decisive factor and plays a pivotal role in establishing client trust when conducting online transactions. This also relates to customers’ confidence in their counterparty’s trustworthiness, ensuring that they will not engage in deception, purposefully harm customer privacy, or exploit their data for alternative purposes [14,59]. People are expected to believe that firms are capable of constructing a platform with a resilient system that possesses both high levels of security and trust. Therefore, the researcher proposes the following:

**Hypothesis** **5** **(H5).**
*WLQ positively influences OGPI.*


### 3.3. Moderating Roles of Green Consumers

#### 3.3.1. Dark-Green Consumer

Dark-green consumers refer to consumers possessing the highest level of green consumption, and their motivation to actively seek information and purchase environmentally friendly products and services comes from their inner intentions [20,60]. Their knowledge about green consumption influences their strong intentions toward green consumption and leads to their decision to pursue environmental interests. This group is willing to pay for premium products while considering environmental impact [61]. Previous studies from Roberts and Bacon (1997) [62] and Zeynalova and Namazova (2022) [63] mentioned that people with high awareness of environmentally friendly consumption are relatively young, better educated, have higher incomes, and mainly consist of women. This is because women have a relatively high commitment to environmental health risks compared to men [64,65]. Apart from that, research showed that the preferences of people with high incomes are different from those with lower incomes in paying attention to green consumption, and people with good education have relatively high preferences for environmentally friendly consumption [63]. Accordingly, it interpreted that dark-green consumers possessed a solid commitment to the environment.

The younger generation is tech-savvy and quickly becoming technology adopters. Previous research studied factors influencing customers’ purchase intentions on the Fresh e-commerce platform [15]. They implemented the leading UTAUT theory and extended the construct of perceived risk to explore the factors that made customers purchase fresh foods online. It revealed that most people with a higher intention to purchase fresh foods from this platform are young adults because they are familiar with and have experience with online shopping. They believed that Performance Expectancy and Social Influence can positively influence their online purchase intention, while perceived risk is vice versa. Therefore, the researcher suggests the following:

**Hypothesis** **6A** **(H6A).**
*PE positively influences dark-green consumers’ OGPI.*


**Hypothesis** **6B** **(H6B).**
*EE positively influences dark-green consumers’ OGPI.*


**Hypothesis** **6C** **(H6C).**
*FC positively influences dark-green consumers’ OGPI.*


**Hypothesis** **6D** **(H6D).**
*SI positively influences dark-green consumers’ OGPI.*


**Hypothesis** **6E** **(H6E).**
*WLQ positively influences dark-green consumers’ OGPI.*


#### 3.3.2. Semi/Light-Green Consumer

Semi/light-green consumers are known to have moderate environmental awareness and lower intentions to seek information about environmentally friendly products and services compared to dark-green consumers [20,60]. They sometimes decide to buy *green* products, but only sometimes. Their commitment to prioritizing sustainability is smaller than that of the dark-green group. Their purchasing decisions are not only considered from an environmental perspective; several factors, such as price and other benefits, may influence them. They may be willing to pay a premium for environmentally friendly products, but they do not always prioritize them over other considerations [11]. Interestingly, a prior study mentioned that this group is the majority of adherents among green consumer groups [66]. The customers’ demography is not far different from dark-green, but the degree of pro-environmental behavior is different.

Most members of the semi-light consumer group are young adult women [13]. This group has the following three tendencies: to shift from semi/light to dark, to stay light/semi, or to become non-green consumers. Innovative technology can drive consumers to consistently choose green products because it provides accessibility and comprehensive benefits. Naruetharadhol et al. (2023) found that the younger generation intends to use e-commerce platforms to buy ugly vegetables because they are health-conscious and aware of environmental issues [3]. The study revealed that technology features can influence their intention to use technology based on their level of health consciousness. Hence, the researcher proposes the following:

**Hypothesis** **7A** **(H7A).**
*PE positively influences semi/light-green consumers’ OGPI.*


**Hypothesis** **7B** **(H7B).**
*EE positively influences semi/light-green consumers’ OGPI.*


**Hypothesis** **7C** **(H7C).**
*FC positively influences semi/light-green consumers’ OGPI.*


**Hypothesis** **7D** **(H7D).**
*SI positively influences semi/light-green consumers’ OGPI.*


**Hypothesis** **7E** **(H7E).**
*WLQ positively influences semi/light-green consumers’ OGPI.*


#### 3.3.3. Non-Green Consumer

Non-green consumers are least aware of the environment and rarely buy and consume environmentally friendly products or services [20,60]. Afridi et al. (2021) and Borau et al. (2021) mentioned that this group mostly belongs to men because stereotypes about green consumption are associated with femininity [67,68]. Then, consumers in this group mostly purchase green products because of accidents or other unforeseen events. However, non-green consumers may possibly purchase products with environmental considerations. Susanty et al. (2022) revealed that they have the possibility of increasing their green consumption closely to that of the semi/light-green group [13]. By making products accessible to consumers without making them exert much effort, the degree of green consumption can be increased [67]. 

E-commerce allows customers to search for environmentally friendly products easily. Ahmad and Zhang (2020) and Naruetharadhol et al. (2023) stated that technology would be helpful if comprehensive infrastructure provided by firms were readily available [3,53]. Consumers believe that technology will be useful and valuable if the necessary infrastructure is available and can provide comfort and convenience [54,69,70]. Therefore, the usefulness of technology can encourage non-green consumers to buy environmentally friendly products as long as they can experience the benefits. Then, the researcher proposes the following:

**Hypothesis** **8A** **(H8A).**
*PE positively influences non-green consumers’ OGPI.*


**Hypothesis** **8B** **(H8B).**
*EE positively influences non-green consumers’ OGPI.*


**Hypothesis** **8C** **(H8C).**
*FC positively influences non-green consumers’ OGPI.*


**Hypothesis** **8D** **(H8D).**
*SI positively influences non-green consumers’ OGPI.*


**Hypothesis** **8E** **(H8E).**
*WLQ positively influences non-green consumers’ OGPI.*


## 4. Research Methodology

### 4.1. Research Model

Figure 2 depicts the proposed framework of this study, sequenced as H1, H2, H3, H4, H5, H6A to E, H7A to E, and H8A to E. This study explored various factors influencing OGPI for imperfect organic fruits and vegetables through an e-commerce platform. The proposed framework adopted factors from UTAUT’s theory and an extended construct of WLQ, considering it an essential factor influencing green consumers’ OGPI e-commerce [58,71]. The UTAUT theory explains individuals’ intentional usage of technology and their purpose in engaging in online purchasing activities for green products. Moreover, in order to delve deeper into individuals’ comprehensive understanding of green consumers’ intentions regarding buying imperfect organic fruits and vegetables via e-commerce, an additional factor such as WLQ has also been considered. This construct is expected to demonstrate how consumers perceive the quality of labels or information displayed on e-commerce platforms. In total, five constructs (PE, EE, FC, SI, and WLQ) were investigated. Moreover, this conceptual framework is moderated by dark-green as the greenest consumer group, semi/light-green as moderate green, and non-green as the least environmentally friendly consumer group.

### 4.2. Data Collection

In the context of consumer research, particularly e-commerce, the population size was unknown as limitless customers were accessing the online platform [72]. Applying the unknown population, calculator.net was run to calculate the sample size and set a minimum margin error of approximately 1% and a confidence level of 99% [73]. The recommendation was to have at least 668 participants. Prior research recommended a minimum sample size of 200 participants for conducting Structural Equation Modeling (SEM) analysis [74]. Therefore, the researcher aimed to collect data from approximately 700 participants.

The data were collected online utilizing a quantitative approach. This study employed a purposive sampling technique, in which researchers deliberately picked population elements based on assessments [75]. Researchers employed an online survey to collect responses from the participants. Then, they distributed the questionnaires (Appendix A) via the internet to diverse communities in Thailand, with a specific focus on individuals who have a keen interest in green consumption and pro-environmental behavior [76]. This scenario was effective because it offered accessibility to cover geographically widespread green consumers, targeting certain behaviors. Additionally, it provided flexibility in terms of the time needed to complete the task, resulting in enhanced data accuracy [76,77,78].

After distributing the questionnaires online, we successfully collected 700 responses. Then, nonconforming responses were eliminated due to incomplete-returned answers from the participants, and 668 data remained. As a result, the usable data rate achieved a level of 95.5%, while the invalid data rate accounted for 4.45%.

### 4.3. Measurement Items

At the beginning of the questionnaire, willing participants were provided with information about the study’s objective, terms, and conditions and the questions section when they opened the first page. The surveyor informed them that the recorded answers and identities used were confidential and kept them anonymous. They had the right to decline if they were unwilling to fill it. Furthermore, the criteria were that they must be at least 18 years old, comprehend the language, and agree to answer all the questions sequentially, which took around 5–10 min. Next, they could move to the first section and answer two interrogative questions. They could fill out the next section when these two questions were completed. These filter questions aimed to improve the quality of the data by removing bias. Accordingly, the Khon Kaen University Ethics Committee for Human Research approved this research under the code HE663190.

Afterward, the second section requested that the participants explain their demographic information. In the third section, the questions were related to the greenest level of consumers, derived from a previous study [63,67]. According to Table 2, it involved 8 questions and 9 linear scales: 1 = strongly disagree, 2 = disagree, 3 = moderately disagree, 4 = slightly disagree, 5 = neutral, 6 = slightly agree, 7 = moderately agree, 8 = agree, and 9 = strongly agree. The following section examines the OGPI constructs adopted from UTAUT theory and an extended construct of WLQ. Referring to Table 3, the participants would answer it by filling out the five-linear scales, representing 1 as strong disagreement, 3 as neutral, and 5 as strong agreement. The higher score indicated excellent tendencies toward green consumption, and vice versa.

### 4.4. Data Analysis

At the beginning of this study, the researcher applied Common Method Variance (CMV) to test the inherent subjectivity involved in the process of selecting respondents and the potential for data bias [80,81]. Harman’s single-factor analysis was performed to confirm that the collected data did not exhibit such issues [82]. The factor extracted a total variance of 32.063%, which was below the threshold of 50% [83]. This indicated that there were no issues with CMV. Next, the multivariate normality test for consumer responses was conducted before running the SEM. The data set, including all constructs, was tested, resulting in skewness (−1.2 to 0.13) and kurtosis (−0.85 to 2.8). According to Chen (2012), which was also recently cited by Dandis et al. (2022), when the skewness and kurtosis values are not greater than 3.0 and 8.0, then they are within the acceptable range to confirm multivariate normality [84,85].

There were steps to analyze the data thoroughly. Firstly, the researcher conducted a multivariate cluster analysis to categorize the green consumer into three groups: dark-green, semi/light green, and non-green consumers [3,21]. Then, crosstabulation was conducted to exhibit the demographic profile in the descriptive statistics of those three groups [86]. Next, a means comparison of green customers was tested using the scores of three segments in order to execute a normality and homogeneity test [87]. The normality test resulted in the Skewness and Kurtosis test of green consumers scores ranging from −0.095 to −0.460, significantly passing the criterion (±2) for a large sample size [21,88,89]. The data set met the normality assumption. Following that, the Levene statistic was employed to assess the homogeneity of the dataset. It revealed that four out of eight scores from green consumers were not statistically significant (*p* > 0.05), thereby failing to conform to the homogeneity of variance for the three distinct groups of green consumers [90]. Then, the Welch ANOVA test was advised when the assumption of equal group means was deemed unacceptable [3,90]. Additionally, this study seeks to find the segment with the highest OGPI scores. This research conducted a one-sample t-test, comparing the OGPI scores of each segment with “4” (referred to as a high level of purchase intention) with a significance level of 95%.

Secondly, this research utilized a measurement model test to observe the credibility of the indicators of each variable by validating the CFA. This stage predicts whether the proposed model is reliable and valid by investigating the God of Fit (GOF), convergent validity, and discriminant validity. The GOF index assesses the discrepancy between the variance-covariance matrix derived from the empirical sample and the model variance-covariance matrix constructed from the modeled construct’s measurements [91,92]. Then, convergent validity involves examining the connections between question statements and latent variables by assessing loadings and cross-loadings, while discriminant validity refers to the degree to which a measurement truly captures a unique concept and is not only a reflection of other related constructs [93,94].

Thirdly, the Structural Equation Model (SEM) to analyze the obtained data, a statistical technique used to measure and analyze the relationship between variables [24,87,95]. Statistical software like AMOS and SPSS were used to assist with this data analysis.

Ultimately, the final step involved examining the multigroup analysis to identify the moderating influence of each segment on the structural relationship [21,96]. A multigroup moderation analysis was conducted to examine the impact of each segment on the whole structural equation model [97]. In order to do this, the concept of measurement invariance (MI) was utilized, with three groups performing as moderators: dark-green, semi/light green, and non-green consumers. MI evaluated the degree to which a psychological measure remained consistent and equivalent across different groups or across time. MI included configural invariance, metric invariance, and scalar invariance, which were utilized to evaluate model stability [98]. The critical ratio for path differences, based on Byrne (2010), was calculated using the MGA technique to assess the factor loadings of the models across two groups, given the threshold of 1.96 [41,96,99]. The following section discusses the research results and analysis.

## 5. Result

### 5.1. Cluster Analysis

As demonstrated by cluster analysis results, the descriptive statics of study segments from Table 4 present three groups of green consumers, consisting of dark-green (n = 225), semi/light-green (n = 241), and non-green (n = 202). The chi-square tests were conducted in which eight of nine tests were significant (<0.01) excluding marital status (*p*-value > 0.01).

Most dark-green consumers (segment 1) are the second highest respondents among the three groups (34.14%) and are predominantly female, reaching up to 24% compared to males (9.7%). Milovanov (2016), Susanty et al. (2022), and Wang et al. (2020) confirmed that women are perceived as being more aware and concerned about environmental issues compared to men [13,100,101]. They are mostly between 18 and 34 years old and have four family members or more. This finding aligned with the previous study which found that young people (18–30) were greener than other age groups [101]. These reasons include being digital natives, being better with technology, and having more information on environmental and sustainable practices [102]. Peers and social networks, especially environmental activism and awareness platforms on social media, affect young people. They might acquire greener behaviors and make green purchases due to positive peer pressure and social expectations [103]. Furthermore, the majority of the consumers belonging to this group are students, followed by business owners, and earn income between 10,001 and 20,000. The dark-green consumers are highly educated, with most respondents currently studying or holding a bachelor’s degree. They show a strong interest in purchasing green food and are passionate about social networking services.

Segment 2, known as semi/light-green consumers, appears to be the most favored group among other segments, with a percentage of 36.5%. This finding aligned with the research from Teresa Foti et al. (2022) [40]. The majority of individuals in this category are females (23.7%) who fall between the age range of 18 and 34 years old. Typically, their family consists of three to four individuals, and they have a monthly income between THB 10,001 and 20,000. Likewise, the previous group, the semi/light-green consumers, predominantly consists of highly educated individuals, including students or those graduating with bachelor’s degrees. Also, they show increased interest in purchasing green food products and regularly using the internet.

Non-green consumers, known as Segment 3, have the smallest member size compared to the other two groups of green consumers, accounting for only 30.65% of the total. Notably, the majority of individuals in this category are males. As aforementioned, males are likely less environmentally conscious than females [40,101,104]. Their age range often falls between 25 and 34 years. Like segments one and two, the majority of individuals in this group consist of four family members, with an income ranging from THB 10,001 to 20,000, and possess a bachelor’s degree. Furthermore, they primarily work as private sector employees who exhibited a lack of willingness to purchase green food consistently despite remaining active users of the internet.

This research performed Welch’s ANOVA to understand the mean differences between the three divided segments with unequal measurements [3,105]. Table 5 presents the means of three different green consumer groups, resulting in *p*-values less than 0.001. The results indicated that the cluster analysis successfully grouped the respondents according to their green level (dark-green, semi/light green, and non-green). The number of means showed significant differences among the three segments. Dark-green consumers had means ranging from 7.45 to 8.42, while semi/light-green consumers started from 5.45 to 6.06, and non-green consumers were between 3.86 and 4.25. Apparently, Segment 1 and Segment 2 presented high GC mean scores and were expected to consume products that have less harmful effects on people and the environment. On the other hand, Segment 3 presents a low consciousness of green products and services.

To obtain more specific information regarding the propensity for OGPI, we conducted a one-sample t-test with a level of significance of 95%. This test allowed us to analyze the variations in means and determine if the sample means were significantly different from the predetermined population mean for each group in relation to OGPI [106]. The predeterminant population mean was filled with four out of five, indicating a high degree of intention to make online green purchases [107,108]. We conducted hypothesis tests using H_0_: µ_OGPI_ ≤ 4 and H_a_: µ_OGPI_ > 4. Rejection of H_0_ indicates a high level of online green purchase intention. The critical value of t > 1.962 was used, given the degree of freedom (df) of 667 (*df* = n − 1 = 668 − 1) and the significance one-tail of 0.05.

According to Table 6, the results suggested that dark-green consumers have a mean value greater than 4, as evidenced by the t-value and one-sided *p*-value. In Segment 2, the results obtained from OGPI1 to OGPI3 indicated that the t-values are greater than 1.962 and the *p*-values are lower than 0.05. is positive, suggesting that the means exceed the hypothesized population mean of 4 [109]. However, the t-values of Segments 2 and 3 are all below 1.962, inferring that the means of these segments are below 4. Hence, only the dark-green consumer segment can be inferred as the potential target segment for imperfect fruits and vegetables online due to high OGPI scores. In addition, the semi/light-green and non-green segments are not considered potential customers.

Consequently, dark-green consumers showed the highest intention to make online green purchases. This indicates they are the most likely buyers or target customers for a new e-commerce platform that sells imperfect organic fruits and vegetables.

Next, the following sections evaluate the measurement model using Confirmatory Factor Analysis (CFA) and Structural Equation Modeling (SEM).

### 5.2. Measurement Model

CFA was used in the measurement model to explore reliability, internal consistency, discriminant validity, and convergent validity [87]. It was carried out by connecting line covariances to constructs. The CFA confirmed the association between the constructs, including the goodness of fit (GOF), average variances extracted (AVE), composite reliability (CR), and Heterotrait–Monotrait (HTMT) correlation ratio. The goodness of fit (GOF) of the link could be increased based on covariances among errors within the same construct [92]. The result revealed all of the measurement model’s goodness of fit was accepted and passed the criterion (Byrne, 2024): CMIN/Df < 3.00; Tucker–Lewis Sustainability Index (TLI), Comparative Fit Index (CFI), Incremental Fit Index (IFI) > 0.90; and Root Mean Square Error Approximation (RMSEA) <0.10. Consequently, CMIN/df (1.922), TLI (0.967), CFI (0.971), IFI (0.974), and RMSEA (0.044) reveal satisfactory GOF results (see Appendix B: Table A1).

Referring to Table 7, this measurement model’s results with the required threshold values for the fit index were utilized to evaluate the Convergent Validity. Cronbach’s Alphas, AVE, and CR were required to measure the degree of consistency. Cronbach’s Alpha determined whether the Likert scale surveys were trustworthy, AVE measured the variation in a construct related to error measurements, and CR explained a construct’s internal consistency and dependability. According to the predetermined thresholds, the values were supposed to be against Cronbach’s Alpha of 0.70, AVE of 0.50, and CR of 0.70, respectively [97]. As the construct’s Convergent Validity outcomes, the scores of PE, EE, FC, SI, WLQ, and OGPI had fully achieved Convergent Validity criteria. Regarding the constructs, all the *p*-values have been achieved significantly. Then, Cronbach Alpha values surpassed the standard of 0.70, while AVE reached higher than 0.50. Moreover, WLQ also successfully met the criteria (>0.70) [110].

Discriminant validity was assessed to determine if the presented constructs exhibited empirical differences. The HTMT analysis was selected as the preferred technique for analyzing discriminant validity due to its precise measurements [97]. Previous research has shown that HTMT exhibited more precise results in addressing collinearity issues among the given constructs [110]. The level of precision achieved was 97–99%, which is significantly higher than the Fornell and Larcker Criterion’s accuracy of 20.82% [21,111]. Therefore, it can be inferred that the HTMT ratio technique is effective in preventing inaccurate analysis for this measurement model. The HTMT values were expected to be less than 0.85 [112]. When the results surpassed the expected values, the discriminant validity was invalid. All the results were lower than 0.85, indicating they were all satisfied (see Appendix B: Table A2).

### 5.3. Structural Equation Model

After completing the reliability and validity tests, the constructs were performed using SEM analysis. All GOF indices pass all the thresholds, for instance, CMIN/df (1.922), TLI (0.967), CFI (0.971), IFI (0.971), and RMSEA (0.037) (see Appendix B: Table A3). Table 8 shows the results of the structural equation model. Four out of five constructs were supported with a significant level of less than 0.05, consisting of PE, FC, SI, and WLQ. Meanwhile, EE was rejected.

H1 was supported, in which PE positively influenced OGPI on e-commerce. Their OGPI improved when they knew the benefits, such as technical features, could help accomplish their goals and environmental impacts [1,48,113]. For instance, buyers may view eco-friendly consumption as more efficient if they can easily find eco-friendly products through platform features that provide clear information, extensive filters, and advanced search options. These features aid in time management and enhance overall productivity [114]. The results were explicitly explained with a standardized estimate of 0.237 and a *p*-value of less than 0.001.

In contrast, H2 was rejected, explaining that EE was insignificant, indicating that there was no relationship between EE and OGPI in e-commerce. The standardized estimate reached an unexpected outcome of 0.021, and the *p*-value was 0.811, surpassing the accepted significant levels. The result infers that customers did not intend to put effort into online purchasing of imperfect fruits and vegetables [23].

Next, H3 was supported, in which FC positively influenced green consumers’ OGPI toward e-commerce. The standardized estimate of 0.265 and *p*-value of 0.003 were accepted. The green consumers had concerns about the relevant facilities and tangible and intangible resources provided by the firm, which can support and maximally achieve the benefits of purchasing fruits and vegetables through the platform [52,53,54]. FC facilitates a reduction in obstacles and simplifies the process of obtaining green products and information, ultimately resulting in a more convenient adoption of sustainable consumption practices [115,116]. E-commerce platforms have the potential to enable consumers to embrace a sustainable lifestyle and make a good environmental contribution by offering comprehensive facilities [115].

Moreover, H4 was supported, in which SI positively impacts OGPI. The standardized estimate of 0.153 and *p*-value of 0.009 were accepted. Advanced technology, such as the internet, quickly spreads information, leading to “zero distance” between the customers and the information [51]. The presence of an online environment that fosters support enables individuals to establish connections, exchange information, and cooperate in the pursuit of sustainability [24,117]. Coworkers, relatives, friends, and online comments can also influence customers to purchase imperfect organic fruits and vegetables via e-commerce if approved or encouraged, resulting in adopting environmentally friendly consumption [3,118]. Therefore, SI plays an important role in fostering a collective commitment to environmental conservation and encouraging green consumption behavior.

H5 was supported. As for the novel construct, WLQ has positively influenced a customer’s OGPI when buying imperfect fruits and vegetables through e-commerce. The result showed a standardized estimate of 0.128 and a *p*-value of 0.025. Green consumers believe that e-commerce is trustworthy and capable of making online green purchases. The perception of a platform’s credibility promotes trust and ethical behavior, which in turn creates an environment where customers have confidence in new e-commerce capabilities [14,17,58,59]. This confidence can encourage customers to make green purchasing transactions in new e-commerce, such as purchasing imperfect organic vegetables through e-commerce to reduce food waste.

### 5.4. Multigroup Moderation Analysis

#### 5.4.1. Measurement Invariance

Measurement invariance (MI) is a method employing indicators to examine latent characteristics among three groups of green consumers [119]. The CFA was performed to assess this information [120]. MI is accepted when the configuration, metric, and scalar invariance criteria are met. Table 9 revealed that the three segments of green consumers are different, respectively.

The measurement invariance test resulted in all satisfaction. CMIN/df showed values less than 1.7, indicating that the result is satisfied. Additionally, all the following fit indices reached the standard criteria for configural invariance, metric invariance, and scalar invariance. TLI, CFI, and IFI were greater than 0.90, and RMSEA was under 0.10, resulting in full measurement invariance [96]. After conducting the measurement invariance test, the following measurements must be performed:

The GOF of the multigroup structural model was performed and brought satisfactory results. The fit index’ values passed the threshold value, leading to an acceptable outcome. CMIN/df (1.588) was less than 3.00, while TLI (0.935), CFI (0.944), and IFI (0.944) presented more than 0.90. Additionally, RMSEA (0.030) met the standard criteria [98].

#### 5.4.2. Z-Test for Loading Differences

The Z-Test for loading differences was conducted to understand the factor loading differences among the three segments: dark-green, semi/light-green, and non-green. It referred to comparing loading differences in terms of magnitude and direction between two loadings in the main structural model [93,121]. The correlation coefficient for variables and factors is basically what factor loading is. The variable’s high loading led to a strong relationship with the factor, and vice versa. Therefore, comparing the two loading groups on a particular variable can demonstrate a stronger, more significant difference between the two segments. This study applied the critical ratio differences of the MGA approach to assess the loadings between the two groups [41]. The critical ratio differences are statistically significant when two loadings for each group differ. Particularly, if the critical ratio’s value is greater than the standard criteria (1.96), the two observed groups’ factor loadings are significantly different [111].

Table 10 compares the factor loadings between the three customer segments: dark vs. semi/light greens, dark vs. non-greens, and semi vs. non-greens. The results showed that only H3 was statistically significant for dark-green vs. non-green consumers. The critical ratio difference met the standard criteria of being higher than 1.96 (2.373 > 1.96) [122]. It was interpreted that non-green consumers express concerns regarding FC, while dark-green consumers believe that, besides FC, other essential factors influence OGPI via e-commerce. This situation may arise due to varying degrees of consciousness. Dark-green consumers possess a heightened awareness of consuming things that are environmentally friendly [13,20,100,104]. They may proactively pursue green choices, sustainable methods, and inclusive communities, irrespective of external motivations or assistance. As a result, they began contemplating ways to decrease their need for additional facilitating conditions. Meanwhile, non-green consumers may require more understanding and improved access to resources to overcome obstacles and embrace sustainable practices. These findings are consistent with the segmentation concept proposed by Polonsky (1995) [20]. Nevertheless, the remaining results for loading differences were not statistically significant (<1.96).

## 6. Discussion

This research work leans upon the UTAUT theory from Venkatesh et al. (2003) in revealing the significant relationships among the factors influencing customers’ OGPI and the mediating role of three groups of green consumers conceptualized by Polonsky (1995) [20,42]. The UTAUT theory is a unified theory formed from antecedent theories associated with behavioral intention and technology adoption. Meanwhile, Polonsky’s segmentation concept classifies and describes green consumers into three groups (dark-green, semi/light-green, and non-green) based on their behaviors.

This current study combines UTAUT theory with an extended construct (WLQ) and green consumer segmentation. This collaboration expands the investigation of green customers’ intention to purchase through e-commerce, particularly imperfect organic fruits and vegetables. An additional construct provides more understanding to look deeper into customers’ perspectives toward the emerging platform. Integrating the UTAUT theory with a new construct and green consumer segmentation concept contributes to theoretical development that can be used to foresee what factors and segments can promote OGPI. Moreover, further explanations of the behaviors and perspectives of three groups of green consumers would contribute to the existing literature and could be applied to future research in many contexts.

### General Behaviors of Dark, Semi/Light, and Non-Greens and Their Online Green Purchase Intention (OGPI)

This part discussed green consumer profiling and multigroup analysis for three customer segments: I. dark-green, II. semi/light green, and III. non-green. PE has positively influenced dark-green and semi/light-green consumers’ OGPI of imperfect organic fruits and vegetables through e-commerce. Additionally, WLQ and SI had statistically significant relationships with OGPI for dark-green consumers only. However, non-green consumers have been positively influenced by FC.

Figure 3 displays the results of three distinct green consumer behaviors. Firstly, the dark-green consumers tend to be mostly female. This finding was relevant to Brough et al. (2016), in which green consumers were stereotyped as more feminine even when the level of greenness and femininity were linked [123]. The findings are consistent with those of Solvalier (2010) and Polonsky (1995), who stated that dark-green consumers were highly environmentally conscious [20,60]. They would love to actively seek green information from various sources, including the internet, related to eco-friendly consumption, putting knowledge before the decision. This segment involves primarily young adults with good educational backgrounds, enrolling for bachelor’s degrees, and living with their families. Apparently, young people were active internet users who loved to see health-related content on different platforms [124]. Their habits were influenced by social media for health and lifestyle inspiration.

Meanwhile, semi/light-green consumers were considered moderately green. This group characteristic was seemingly dark-green, excluding their family size, but the degree of greenest is lower than theirs [20,60,123,124]. From 1 to 9, the semi/light group obtains approximately 5.49 to 6.06 scores. They were environmentally conscious. They deliberately buy green foods or environmental products, but it only happens occasionally. Some conditions, such as costs and green marketing issues, can influence their decision to consume green products.

In contrast, non-green consumer behaviors differed greatly from the previous two segments. Most of the members were males, which was aligned with the previous finding [123]. This group graduated with bachelor’s degrees, lived with family members, and worked for private companies. Unfortunately, they presented negative attitudes toward green behaviors [125]. Non-green consumers did not pay any attention to seeking green products or foods for the sake of protecting the environment. Their awareness level was around 3.68 to 4.25, which was the lowest level of green consumers compared to others. Accordingly, this group was called low environmentally conscious. Green consumption was found to be an accident [20,60].

Multi-group analysis revealed the cross-sectional relationships for the constructs. According to Figure 4 and Figure 5, PE revealed that it had positively influenced the OGPI of dark-green (Std.Est. = 0.206, *p* = 0.21) and the OGPI of semi/light-green (Std.Est. = 0.385, *p*= 0.007). As mentioned, these groups are considered digital-native and are in the techno-savvy generation [102]. These groups believe that utilizing technology with advanced technical features can support their daily lives to be more effective and efficient regarding their time and performance [3,115]. Notably, e-commerce easily creates transactions on any occasion where customers need to operate an online shopping channel and purchase things online [126]. E-commerce is a reliable shortcut to shorten their shopping time. Additionally, dark-green and semi/light-green consumers are willing to contribute to sustainability activities by adopting e-commerce [127]. They are pro-environmental awareness [31]. These two green groups believe e-commerce can be applied to build sustainable food and reduce food waste [128]. Their willingness to use technology and their environmental awareness may guide them to purchase imperfect organic fruits and vegetables through a circular economy-based e-commerce platform. Unfortunately, non-green consumers do not seem to recognize the advantages of PE in utilizing the online channel (see Figure 6). When consumers realize the merits of using E-commerce in shopping and possess satisfying experiences with e-commerce, their purchase intention will increase [129].

Interestingly, dark-green’s OGPI was positively affected by SI (Std. Est. = 0.209, *p* = 0.038) (see Figure 4). As aforementioned, the presence of behaviors, the presence of internet access, and social expectations frequently have an impact on OGPI [21,24,51,115]. These people believed that family members and close friends could encourage them to be sustainable green consumers and recommend trusted information with proper publicity [115,130]. They place trust in and accept the advice of a substantial network of friends, family members, or other individuals who support the importance of online transactions [40]. Decision-making also includes the process of shaping customers’ future intentions. This finding is aligned with previous research in which people’s willingness to pay can be affected by SI [3]. Improving the e-commerce platform in the context of an impressive image is crucial because it can reveal potential ways for peers to influence their surroundings [115]. Hence, SI favorably affects dark-green consumers’ decisions when actively searching for green products.

Additionally, WLQ (novel construct) has positively influenced the OGPI of dark-green consumers (Std.Est. = 0.170, *p*= 0.043) (see Figure 4). Digital label credibility, encompassing visual representation, data accuracy, reliability, and comprehensibility, is a matter of concern for this group and is in line with prior findings. Because of these things, they have confidence in the information supplied on the website and are comfortable buying imperfect organic produce. Based on the basic evaluation of the platform, they may determine the platform’s credibility and reliability [58,59]. If the platform consistently delivers high-quality content and is technically stable, users might feel more confident using it. Consumers have more assurance when making purchases online, which elevates their level of trust. Therefore, enhancing WLQ is very important in influencing dark-green consumers’ OGPI.

In contrast, Figure 6 reveals that FC positively influences the OGPI of non-green consumers (Std. Est. = 0.5777, *p* = ***). They believed that technology would be advantageous and valuable if the necessary infrastructure were readily available [3,53]. Having the appropriate tools to fulfill essential requirements for utilizing online shopping platforms in a sustainable manner, such as possessing mobile internet connectivity, possessing the necessary knowledge and expertise, and receiving help from relevant parties, all contribute to increasing individuals’ propensity to engage in e-commerce [69,70]. The online platform serves as a crucial catalyst for online green purchase transactions, as highlighted by Le (2021) and Widyanto et al. (2021) [54,69]. These factors may be linked to the user’s convenience when utilizing technology. The extensive range of amenities enables them to reap the primary advantages of e-commerce, hence reducing the reluctance of non-green consumers to engage in e-commerce transactions. In line with prior research, the existence of extensive infrastructure and technical elements can influence individuals’ online green purchasing intention as a result of their perception of convenience [3,52,53]. Non-green consumers believe that FC plays a significant role in their decision-making process when selecting green products through e-commerce platforms.

Ultimately, EE revealed that it is not significantly influenced by OGPI with dark-green, semi/light-green, and non-green consumers. The MGA results discovered that none of these groups required EE. The findings for EE are statistically aligned with prior studies [23,115]. According to EE, green consumers are not willing to put more effort into purchasing imperfect fruits and vegetables through an emerging online platform [115]. Their purchase intention will increase when e-commerce possesses a higher level of technology readiness [131]. It clearly shows that EE has not become essential for an emerging e-commerce setting, especially green segments.

## 7. Research Implication

This study examined the characteristics and behaviors of three green consumers and their OGPI of imperfect produce via e-commerce using the UTAUT theory. The proposed framework incorporated WLQ as an expanded concept. This research was expected to provide valuable contributions to academic purposes, food policy, e-commerce technical advancement, and business purposes.

### 7.1. Theoretical Contribution

In this study, the researchers addressed the gap between three groups of green consumers and their online green purchase intentions. It significantly contributes to the current literature review on digital marketing and green consumption. First, this research uncovered three varying levels of green consumers based on the level of green consumption, utilizing Polonsky’s (1995) concept [20]. These findings also revealed their behaviors and preferences toward online green purchase intentions via e-commerce. Multigroup analysis was applied, resulting in factors influencing the intention to purchase green products online among dark-green, semi/light green, and non-green consumers, which have been revealed in detail. Furthermore, it confirmed that the findings provide a more comprehensive understanding of the theory’s applicability and segmentation concept and emphasize the nuances of behavioral variations that can impact the intention to make online green purchases through e-commerce platforms.

Secondly, this research advanced the theoretical contribution by creating a new construct, Web-based Label Quality (WLQ). This construct aimed to better understand green consumers’ perceptions regarding the label of the information displayed in e-commerce. WLQ has not been examined in previous research. Therefore, this research novelty lies in expanding perception and influencing purchase intention through a new lens, specifically in the applicability of the e-commerce context. This research found that dark-green consumers considered WLQ to be one of the factors influencing their willingness to buy imperfect organic produce online. The significant result indicated that the questions and hypotheses in this construct were empirically validated. Moreover, integrating a newly added construct and the leading UTAUT theory extended the compatibility of the theory and enhanced conceptual understanding.

### 7.2. Practical Implication

According to OGPI results, dark-green consumers are the main target for selling imperfect organic vegetables through e-commerce. PE, SI, and WLQ were the significant factors influencing their OGPI. Developers of new technologies can take PE and WLQ into account while developing complex e-commerce platforms, giving users a taste of the technology’s capabilities while they shop online. By combining PE with WLQ, they can build features or systems that boost the customer’s performance on a regular basis and increase customer engagement in the platform, which is crucial for online sales. For instance, businesses and developers may adjust to consumers’ needs by incorporating features that improve their day-to-day lives and earn their confidence. Features like easy search, access, and purchase, as well as connection with numerous payment functions, allow for quick, safe, and secure monetary transactions [14,69]. Moreover, Additionally, the developer and designer can collaborate to raise awareness and create e-commerce tactics by tailoring the aesthetic design to the preferences of dark-green consumers, which will help the business achieve its goals, for instance, encouraging customers to purchase and re-purchase, revisit, and find the website [16]. It is highly recommended that more consumers thoroughly utilize the online system to appreciate its benefits compared to traditional shopping channel systems.

In addition, SI was influenced by dark-green-associated people. This situation can be utilized by firms and policymakers for marketing purposes. Dark-green consumers generally place trust in acquaintances, such as family, friends, and coworkers, who have firsthand knowledge and have assessed internet purchases. Although opinions are subjective, they have a considerable impact on individuals who are uncertain about engaging in online green buying. Therefore, firms must strengthen their reputation by obtaining good endorsements from individuals who have personally used their online green purchasing services. On the other hand, policymakers may use SI to promote imperfect organic vegetables in society. They might collaborate with food manufacturers and restaurants to include ugly organic veggies in their products and menu offerings. They can also initiate marketing strategies to enhance consumer perception and acceptance of imperfect organic produce. This can aid in standardizing their consumption. Implementing educational initiatives, such as launching advertisements, can effectively inform consumers about the advantages of imperfect organic produce, emphasizing their nutritional worth and the ecological consequences of food wastage. In order to effectively target a wide range of people, it is suggested that social media and TV commercials be utilized and also partnerships with influential individuals be established. By employing SI, policymakers can encourage grocery stores and marketplaces to promote and label ugly veggies to dark-green consumers, first by emphasizing cost reductions to persuade consumers to prefer these things.

The growth of companies Is significantly influenced by technology, as it has transformed the traditional reliance on human interaction into a more streamlined and convenient process [45]. Consequently, customers increasingly expect the most advantageous outcomes while engaging with internet companies. Consequently, organizations with websites or applications exhibiting suboptimal performance may have significant challenges in effectively adjusting to the prevailing changes in their environment.

## 8. Limitations and Future Research

The study’s limitation was found in the responses of green consumers. Most of the respondents in this study were young people. Considering the equality of responses for each generation can improve future studies. In addition, this study examines the factors that impact three categories of green consumers’ adoption of new e-commerce platforms. The focus of the study is mostly on consumers’ acceptance and perceptions of technology, which in turn affects their willingness to purchase online. Future research should integrate theories on technological adoption and green consumption to investigate potential elements that can improve continuous online transactions associated with the green consumption concept through e-commerce, as opposed to emerging e-commerce. For example, further study may expand the theory by utilizing UTAUT-2 to identify the OGPI of green consumers and green consumption-related constructs to further comprehend the variables and attitudes between green consumers and the continuous OGPI using sustainable e-commerce technology [132].

Moreover, there were available segmentation concepts that could categorize green people specifically according to their own spirits, such as animal lovers, health fanatics, outdoor enthusiasts, and those who despise waste [33]. Researchers can use this concept to determine which set of customers are more inclined to purchase imperfect organic products from e-commerce and to investigate the comprehensive behaviors of green consumers in dynamic future research.

## 9. Conclusions

Consumers’ misperception of imperfect fruits and vegetables leads to these foods contributing to FVW [133,134]. In fact, these fruits and vegetables still contain abundant nutritional content. An emerging e-commerce platform sells misshapen organic vegetables through online scenarios to combat the waste from misshapen organic vegetables. The researchers applied the UTAUT theory to examine the market demand for e-commerce. Then, this study extended the UTAUT model with an extended construct: Web-Based Label Quality (WLQ) [14,58,59]. We employed the cluster analysis method to classify the respondents into dark-green, semi/light green, and non-green consumers. Then, we investigated the relationship among factors to determine the OGPI of each green segment. The respondents to this study were up to 668 people from various demographics. The result revealed that PE, FC, SI, and WLQ positively influence green consumers’ OGPI through e-commerce.

By MGA analysis, the results are revealed based on the preferences of each segment. Dark-green consumers’ OGPI is influenced by three factors: PE, SI, and WLQ. Semi/light-green’s OGPI through e-commerce is only positively affected by PE. Meanwhile, FC positively influences non-green consumers’ OGPI. This study emphasizes the importance of considering three distinct customer groups—dark-green, semi/light-green, and non-green—while examining their inclination to buy imperfect organic fruits and vegetables online via e-commerce. These valuable discoveries may be utilized to enhance platform marketing tactics, technological improvements, and business operations to attract online consumers’ intentions to make green purchases online [3,115,135]. This can be achieved by comprehending the factors that impact green customers’ disposition to make online green purchases. The stakeholders in the organic fruit and vegetable sector in Thailand, who are involved in unsatisfactory production, might extensively utilize this research’s findings.

## Figures and Tables

**Figure 1 foods-13-01401-f001:**
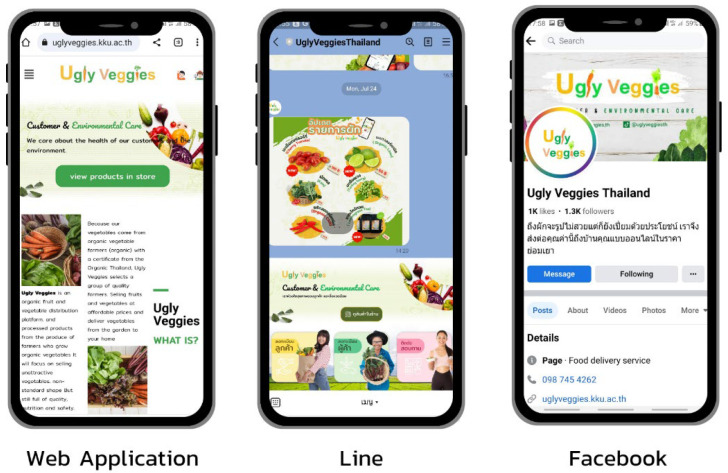
Ugly Veggies Platform. Source: https://uglyveggies.kku.ac.th/ (accessed on 20 September 2023). Note: The Thai language statements are translated into “Even though the vegetables are not beautiful, they are still full of benefits. We can bring this value to your home online at affordable prices”.

**Figure 2 foods-13-01401-f002:**
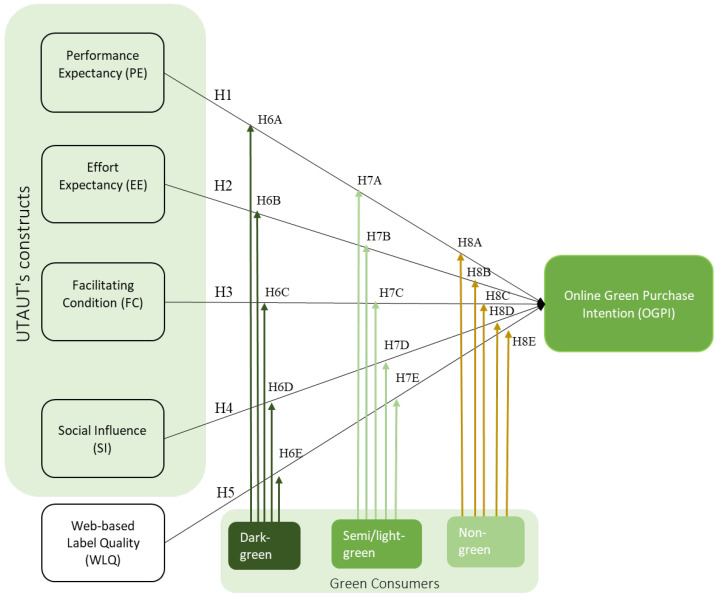
Research model. Source: Figure created by the author, 2024.

**Figure 3 foods-13-01401-f003:**
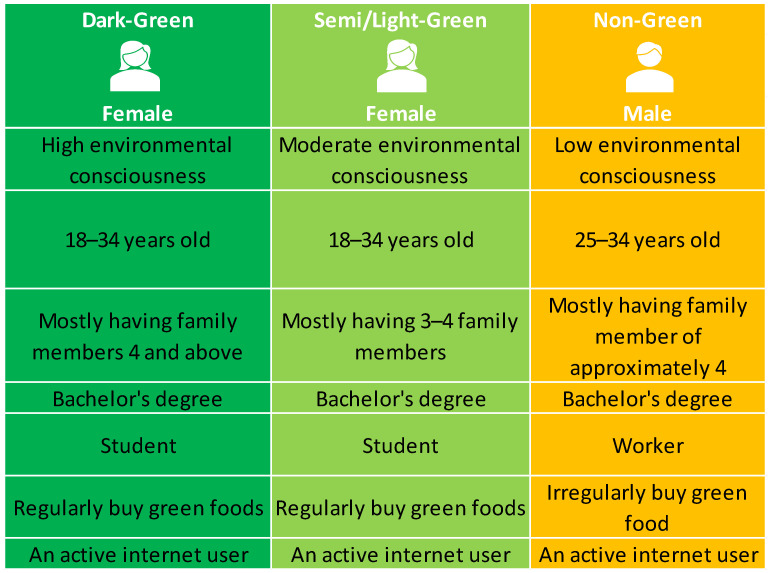
Comparison of consumer behavior based on green segmentation.

**Figure 4 foods-13-01401-f004:**
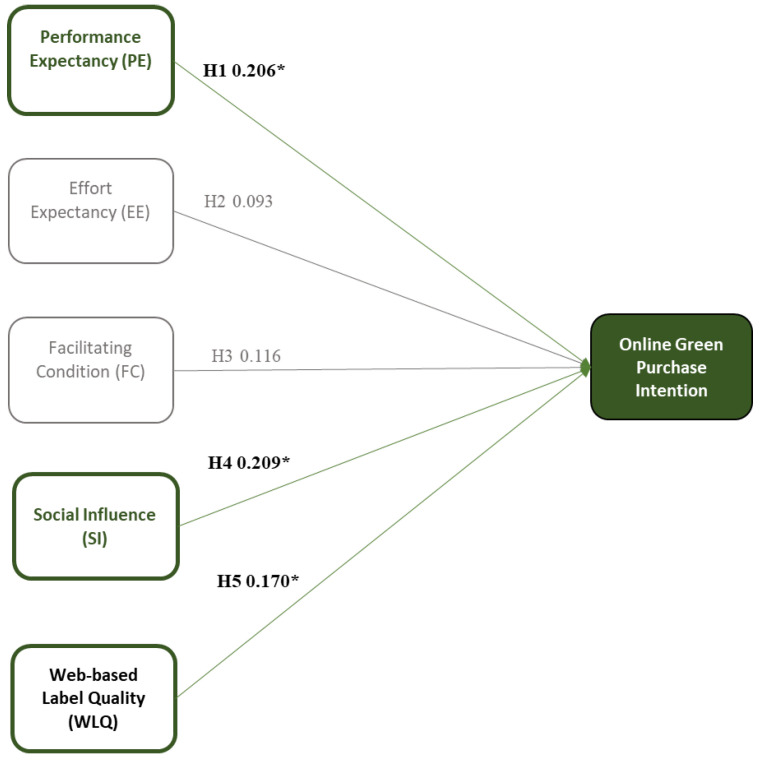
The SEM of dark-green consumers. Note: * at <0.05.

**Figure 5 foods-13-01401-f005:**
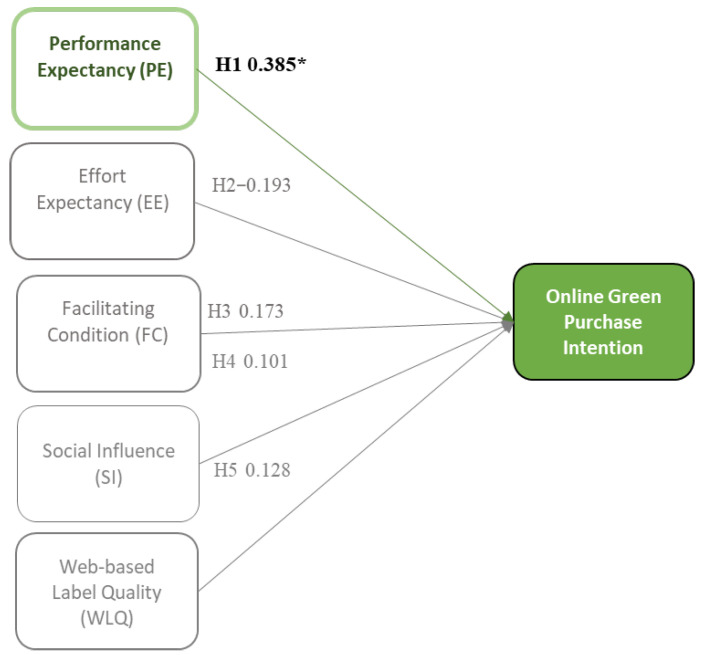
The SEM of semi/light-green consumers. Note: * at <0.05.

**Figure 6 foods-13-01401-f006:**
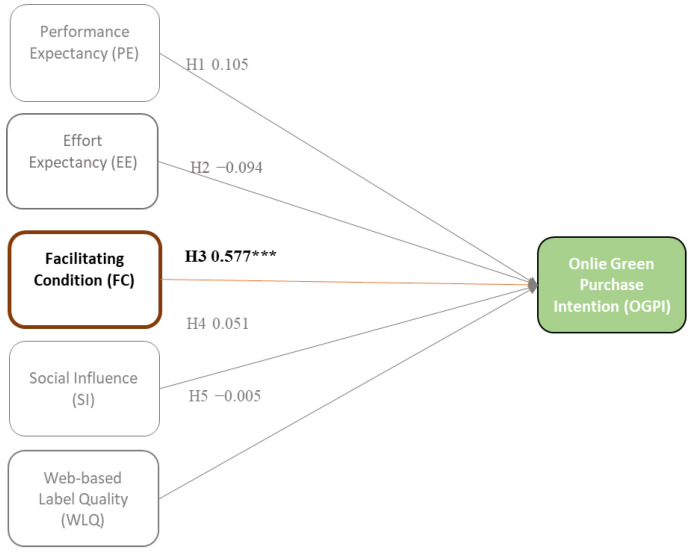
The SEM of non-green consumers. Note: *** denotes significant at <0.001.

**Table 1 foods-13-01401-t001:** Green consumer segmentation concepts from the previous literature.

Segmented by	Characteristics
Polonsky (1995) [20]	Dark-green: Their motivation to actively seek information about green products, services, and purchases is derived from their inner intentions. Semi/Light-green: The intention to seek information on green products and services is lower than that of dark-green consumers. They decide to purchase green products sometimes but not all the time. *Non-green*: Rarely buy and consume green products or services. If they purchase such green products, it unintentionally happens.
Ottman (2010) [33]	Resource conservers (hate waste): They prioritize economic value, long-lasting, and reusability advantages of products. In addition, the products that enable them to recycle, compost, and save energy. Health fanatics: They are concerned about excessive sun exposure, fear of chemicals used in products, and fear of contaminants in children’s toys. They consider organic elements, health benefits, trust, and natural ingredients. Additionally, they encourage cross-promotion with organic food parties, sponsorships, or promotions in natural life magazines. Animal Lovers: People prefer vegetarianism, consider PETA (People for the Ethical Treatment of Animals), and boycott animal exploitation. Also, they seek products that are “cruelty-free.” Outdoor Enthusiasts: People entertain themselves by doing nature tourism, such as camping, mountain climbing, skiing, hiking, and visiting national parks. They are eager to cut the environmental impact of recreational activities. They are additionally concerned with labeled items and recyclable materials.
Organization and Wax (1990) [32]	True Blue Greens (9%): Using strong environmental values to affect change positively, they repeatedly avoid products manufactured by an environmentally unconcerned corporation. Greenbacks Green (6%): Greenbacks differ from True Blues in many ways, including their environmental beliefs. They are, nonetheless, more prone to buy green items than the average green customer. Sprouts (31%): Sprouts believed more in theory than practice. They seldom order a green product if it is more costly. In fact, they can afford eco-friendly products and are willing to use them when people can persuade them the right way. Grousers (19%): Grousers tend to neglect the environment and are cynical about their ability to contribute to it. They think environmentally friendly products are expensive and do not greatly impact product competition. Basic Browns (33%): They are solely concerned with their daily lives and are disinterested in environmental and social issues.

**Table 2 foods-13-01401-t002:** Green consumer segmentation.

Constructs	Item	Observed Variables	Source
Green Consumer	GC1	I tend to buy degradable products that easily blend in with the environment.	[63,67]
	GC2	I prefer to purchase a similar item in a larger package.	
	GC3	I procure used things to cut down on unneeded consumption.	
	GC4	The food that has not been completely consumed is stored, processed, or given to others.	
	GC5	I try not to waste anything in my home.	
	GC6	If I am aware of the possible environmental harm that any products may cause, I will not purchase them.	
	GC7	I usually try to choose items that are environmentally friendly and contain less pollution when shopping.	
	GC8	When offered the option of two similar products, I definitely choose the one that poses the least risk to other people and the environment.	

**Table 3 foods-13-01401-t003:** Constructs and observed variables.

Constructs	Item	Observed Variables	Source
Performance Expectancy (PE)	PE1	E-commerce selling imperfect organic fruits and vegetables has the potential to boost my daily performance.	[42,43,49,69]
	PE2	Buying fruit and vegetables on e-commerce can save me some time.	
	PE3	Purchasing through e-commerce can be accessed from anywhere.	
	PE4	The e-commerce selling imperfect organic fruits and vegetables helps me make purchases more efficiently.	
Effort Expectancy (EE)	EE1	It is simple to figure out how to operate an e-commerce site selling imperfect organic fruits and vegetables.	[42,43,49,69,79]
	EE2	The e-commerce selling imperfect organic fruits and vegetables provides a user-friendly interface.	
	EE3	The e-commerce selling imperfect organic fruits and vegetables is less confusing to adopt	
	EE4	It doesn’t take long to become an expert user who understands e-commerce selling imperfect organic fruits and vegetables.	
Facilitating Condition (FC)	FC1	I own the required resources to use e-commerce selling imperfect organic fruits and vegetables.	[42,43,49,69]
	FC2	I am knowledgeable enough to use e-commerce.	
	FC3	I assume that assistance from the company is available if I have trouble with the platform.	
	FC4	The e-commerce selling imperfect organic fruits and vegetables will function similarly to other e-commerce systems.	
Social Influence (SI)	SI1	I feel that the people around me will recommend using e-commerce to purchase imperfect organic fruits and vegetables online.	[42,43,48,49,69]
	SI2	I believe that those influencing my consumption behavior will advise me to adopt e-commerce selling imperfect organic fruits and vegetables.	
	SI3	People adopting e-commerce selling imperfect organic fruits and vegetables will look more prestigious.	
	SI4	I have feelings that the people closest to me will recommend that I purchase imperfect organic fruits and vegetables from e-commerce.	
Web-Based Label Quality Perception (WLQ)	WLQ1	I have confidence that an e-commerce platform offering information on imperfect organic fruits and vegetables provides trustworthy label quality for making a purchase decision.	[14,15,16,17]
	WLQ2	I believe that stakeholders involved in the online sale of imperfect organic fruits and vegetables provide trustworthy label quality for making a purchase decision.	
	WLQ3	The quality and accuracy of label information on the e-commerce website for imperfect organic fruits and vegetables are sufficient to proceed confidently with a purchase transaction.	
	WLQ4	The e-commerce platform selling imperfect organic fruits and vegetables will continually update and improve label information to ensure the security and trustworthiness of a transaction.	
Online Green Purchase Intention (OGPI)	OGPI1	I plan to use e-commerce to purchase imperfect organic fruits and vegetables in the near future.	[15,43,48,49,53]
	OGPI2	I perceive that I will purchase imperfect organic fruits and vegetables through e-commerce in daily life.	
	OGPI3	I will find myself frequently ordering imperfect organic fruits and vegetables through e-commerce.	

**Table 4 foods-13-01401-t004:** The demographics of respondents (descriptive statistics).

Demographic Variable	Categories	Segment 1 (Dark-Green)		Segment 2 (Semi/Light-Green)		Segment 3 (Non-Green)		Total		Significance Chi-Square Test
		*n*	%	*n*	%	*n*	%	*n*	%	
Segment size		225	34.14	241	36.57	202	30.65	668	100	
Gender	Male	65	9.7	83	12.4	105	15.7	253	37.9	<0.001 *
	Female	160	24	158	23.7	97	14.5	415	62.1	
Status	Single	163	24.4	191	28.6	147	22	501	75	0.232
	Married	56	8.4	45	6.7	53	7.9	154	23.1	
	Divorced	6	0.9	5	0.7	2	0.3	13	1.9	
Family	1	15	2.3	5	0.8	9	1.4	29	4.4	<0.001 *
	2	22	3.3	10	1.5	21	3.2	53	8.1	
	3	34	5.1	74	11.1	44	6.6	152	22.8	
	4	83	12.4	83	12.4	83	12.4	249	37.3	
	>4	71	10.6	69	10.3	45	6.7	185	27.7	
Age	18–24	85	12.7	109	16.3	34	5.1	228	34.1	<0.001 *
	25–34	60	9	78	11.7	72	10.8	210	31.4	
	35–44	49	7.3	32	4.8	43	6.4	124	18.6	
	>44	31	4.6	22	3.3	53	7.9	106	15.9	
Income	<10,000	62	9.3	36	5.4	13	1.9	111	16.6	<0.001 *
	10,001–20,000	75	11.2	134	20.1	64	9.7	274	41	
	20,001–30,000	33	4.9	33	4.9	37	5.5	103	15.4	
	30,001–40,000	23	3.4	18	2.7	48	7.2	89	13.3	
	>40,001	32	4.8	20	3	39	5.8	91	13.6	
Education	Diploma	35	5.2	15	2.2	23	3.4	73	10.9	<0.001 *
	Bachelor	152	22.8	207	31	169	25.3	528	79	
	Graduate	38	5.7	19	2.8	10	1.5	67	10	
Occupation	Student	86	12.9	112	16.8	30	4.5	228	34.1	<0.001 *
	Government	39	5.8	53	7.9	49	7.3	141	21.1	
	State Enterprise	11	1.6	16	2.4	13	1.9	40	6	
	Employee	43	6.4	42	6.3	77	11.5	162	24.3	
	Business Owner	46	6.9	18	2.7	33	4.9	97	14.5	
Regular buyer of green food	Yes	215	32.2	231	34.6	79	11.8	525	78.6	<0.001 *
	No	10	1.5	10	1.5	123	18.4	143	21.4	
An active online user	Yes	222	33.2	236	35.3	168	25.1	626	93.7	<0.001 *
	No	3	0.4	5	0.7	34	5.1	42	6.3	

Note: * denotes *p*-value < 0.001 significance level.

**Table 5 foods-13-01401-t005:** Compare means, Standard Deviation (SD), and Welch’s Anova tests of green consumers.

Measure	Segment 1 (Dark-Green)		Segment 2 (Semi/Light-Green)		Segment 3 (Non-Green)		Welch’s Statistic	*p*-Value
	Mean	SD	Mean	SD	Mean	SD		
GC1	8.13	1.176	5.87	1.118	4.25	0.973	694.23	<0.001 *
GC2	7.67	1.570	5.71	0.920	3.81	0.868	574.508	<0.001 *
GC3	7.45	1.734	5.49	1.159	3.89	1.069	347.280	<0.001 *
GC4	7.73	1.509	5.45	1.264	3.83	0.995	513.505	<0.001 *
GC5	8.18	1.058	5.74	1.115	3.77	0.940	1036.880	<0.001 *
GC6	8.24	0.853	5.80	0.924	3.91	0.950	1253.824	<0.001 *
GC7	8.19	0.997	5.85	0.999	3.88	1.017	983.377	<0.001 *
GC8	8.42	0.746	6.06	1.047	3.68	1.003	1554.542	<0.001 *

Note: * denotes *p*-value < 0.001 significance level.

**Table 6 foods-13-01401-t006:** One sample T-test for analysis of target customers.

Measure	Segment 1 (Dark-Green)			Segment 2 (Semi/Light-Green)			Segment 3 (Non-Green)		
	Mean	t	One-Sided *p*	Mean	t	One-Sided *p*	Mean	t	One-Sided *p*
OGPI1	4.24	4.985 *	<0.001 *	4.04	0.988	0.162	3.92	−1.456	0.147
OGPI2	4.17	3.463 *	<0.001 *	3.98	−0.507	0.306	3.91	−2.989	0.003
OGPI3	4.16	3.310 *	<0.001 *	4.03	0.784	0.217	3.76	−4.115	<0.001

Note: * for t-value denotes t > 1.962 given df between 100 and 1000; and * for one-sided *p* denotes *p*-value < 0.05 significance level.

**Table 7 foods-13-01401-t007:** Convergent validity test results.

Constructs	Indicator	Loading	*p*-Value	Cronbach α	AVE	CR
Performance Expectancy	PE1	0.77	***	0.855	0.596	0.855
	PE2	0.763	***			
	PE3	0.749	***			
	PE4	0.806	***			
Effort Expectancy	EE1	0.832	***	0.884	0.659	0.885
	EE2	0.773	***			
	EE3	0.812	***			
	EE4	0.828	***			
Facilitating Condition	FC1	0.81	***	0.831	0.559	0.835
	FC2	0.699	***			
	FC3	0.738	***			
	FC4	0.739	***			
Social Influence	SI1	0.791	***	0.849	0.595	0.854
	SI2	0.785	***			
	SI3	0.691	***			
	SI4	0.814	***			
Web-Based Label Quality	WLQ1	0.799	***	0.862	0.6159	0.865
	WLQ2	0.725	***			
	WLQ3	0.79	***			
	WLQ4	0.822	***			
Online Green Purchase Intention	OGPI1	0.838	***	0.869	0.689	0.870
	OGPI2	0.854	***			
	OGPI3	0.799	***			

Note: *** denotes significant at <0.001.

**Table 8 foods-13-01401-t008:** The structural model test results.

Path	Relationship	Standardized Estimate	*p*-Value	Result
H1	Performance Expectancy (PE) → Online Green Purchase Intention (OGPI)	0.237	***	Supported
H2	Effort Expectancy (EE) → Online Green Purchase Intention (OGPI)	0.021	0.811	Rejected
H3	Facilitating Condition (FC) → Online Green Purchase Intention (OGPI)	0.265	0.003 **	Supported
H4	Social Influence (SI) → Online Green Purchase Intention (PI)	0.153	0.009 **	Supported
H5	Web-Based Label Quality (WLQ) → Online Green Purchase Intention (OGPI)	0.128	0.025 *	Supported

Note: *** denotes significant at <0.001, ** at 0.01, and * at <0.05.

**Table 9 foods-13-01401-t009:** The measurement invariance (MI) results.

Fit Index	Configural Invariance (Unconstrained)	Metric Invariance (Measurement Weight)	Scalar Invariance (Measurement Intercepts)	Threshold
*p*-value	0.000	0.000	0.000	
CMIN/df	1.597	1.586	1.637	<3.00
TLI	0.934	0.935	0.930	>0.90
CFI	0.943	0.942	0.933	>0.90
IFI	0.943	0.942	0.934	>0.90
RMSEA	0.030	0.030	0.031	<0.10
Assessment	Passed	Passed	Passed	

**Table 10 foods-13-01401-t010:** The loading differences among the three segments.

Hypothesis	Causal Relationship	Dark-Green		Semi/Light Green		Non-Green		Critical Ratio Differences			
		Std. Est.	*p*-Value	Std. Est	*p*-Value	Std. Est	*p*-Value	Dark vs. Semi	Dark vs. Non	Semi vs. Non	Threshold
H1	PE →OGPI	0.206	0.041 *	0.385	0.007 **	0.105	0.450	1.146	−1.548	−1.651	|1.96|
H2	EE →OGPI	0.093	0.389	−0.193	0.512	−0.094	0.605	−0.926	−0.920	0.319	|1.96|
H3	FC →OGPI	0.116	0.317	0.173	0.627	0.577	***	0.241	2.373 *	0.604	|1.96|
H4	SI →OGPI	0.209	0.038 *	0.101	0.157	0.051	0.579	−0.475	−1.131	−0.263	|1.96|
H5	WLQ→OGPI	0.170	0.043 *	0.128	0.421	−0.005	0.960	−0.358	−1.443	−0.707	|1.96|

Note: *** denotes significant at <0.001, ** at 0.01, and * at <0.05.

## Data Availability

The original contributions presented in the study are included in the article, further inquiries can be directed to the corresponding author.

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
