# Peer review of "Green Consumer Profiling and Online Shopping of Imperfect Foods: Extending UTAUT with Web-Based Label Quality for Misshapen Organic Produce"

_foods, 2024, doi:10.3390/foods13091401_

Round 1
Reviewer 1 Report
Comments and Suggestions for Authors
This paper uses SEM and MGA models to explore factors affecting their Online Green Purchase Intention (OGPI) for imperfect organic vegetables based on 668 internet users. The topic is interesting and the methods used are correct. But the contributions and writing logic of the paper are not well. Here are some comments.
1. The authors do not state clearly their contributions to the literature. This should be added either in the introduction section or in the literature review section. Please enhance the academic gap you are going to fill.
2. Section “2. Literature Review and Related Works” is not only Literature Review, but contain the Hypothesis. I suggest the authors to split this section into two sections of “2. Literature Review and Related Works” and “theory analysis and Hypothesis”.
3. Section “3. Research Methodology” is too long and contains many unnecessary information, should be reduced. For example, section 3.2. Data Analysis, the four steps to analyze the data is so long and unnecessary. The authors should only present the basic information, not the detailed one, since the four steps is quite normal and has detailed information in section 4. Result.
4. There are some tables are unnecessary. For example, Table 5. The measurement model’s goodness of fit, Table 6. Convergent Validity test results, Table 7. The discriminant validity test with the HTMT approach, Table 8. The main Structural Equation Model (SEM) goodness of fit, Table 11. The Multigroup structural model’s goodness of fit,. These are only the test of the method. These results are not the main focus of the paper, the authors can mention some data in the manuscript, not showing the whole table results in the manuscript.
Comments on the Quality of English Language
Moderate editing of English language required
Author Response
The authors would like to thank you for your valuable comments. Here are the responses.
Reviewer 1
- The authors do not state clearly their contributions to the literature. This should be added either in the introduction section or in the literature review section. Please enhance the academic gap you are going to fill.
Answer:
The academic gap has been enhanced in both introduction and literature review. Please refer to the highlights on page 2 and 3.
- Section “2. Literature Review and Related Works” is not only Literature Review, but contain the Hypothesis. I suggest the authors to split this section into two sections of “2. Literature Review and Related Works” and “theory analysis and Hypothesis”.
Answer:
This section is split to be “2. Literature Review and Related Works” and “3. Theoretical Framework and Hypothesis Development”. Please refer to page 3 and 6.
- Section “3. Research Methodology” is too long and contains many unnecessary information, should be reduced. For example, section 3.2. Data Analysis, the four steps to analyze the data is so long and unnecessary. The authors should only present the basic information, not the detailed one, since the four steps is quite normal and has detailed information in section 4. Result.
Answer:
Explanations on the methodology, especially on data collection and data analysis, have been shortened. Please see the highlights on pages 14-15. However, other parts in the method section contain essential details specific to this research. The author decided to maintain those parts to allow readers to follow the logic and repeat this process.
- There are some tables are unnecessary. For example, Table 5. The measurement model’s goodness of fit, Table 6. Convergent Validity test results, Table 7. The discriminant validity test with the HTMT approach, Table 8. The main Structural Equation Model (SEM) goodness of fit, Table 11. The Multigroup structural model’s goodness of fit. These are only the test of the method. These results are not the main focus of the paper, the authors can mention some data in the manuscript, not showing the whole table results in the manuscript.
Answer:
Some tables that were deemed not necessary were removed, but the Convergent Validity table were considered to be retained. The reason is that there are Factor Loading values for the variables being measured, Cronbach alpha, Composite Reliability, and AVE in the table to indicate the quality of the measurement instrument as well as the reliability of responses and Construct Validity of the construct being measured.
Reviewer 2 Report
Comments and Suggestions for Authors
Moderating effet of three vegi categories need to be justified by the review of literature. Authors need to present the research hypothese for moderating effect.
Method strcutre might be better using the following structure
: 1. Research model, 2. Data collection 3. Measurement items 4. Analysis
Hypotheses number and direction needs to be presented in the research model.
Ifnpossible, median split analysis for moderating effect is recommended.
Figure presenting results need to be presented separately.
Theoretical contribution needs to be upgraded as more convincing manners.
Author Response
The authors would like to thank you for your valuable comments. Here are our responses.
Reviewer 2
- Moderating effect of three veggie categories need to be justified by the review of literature. Authors need to present the research hypotheses for moderating effect.Answer:
The moderating effects of three veggies and its hypothesis have been added. Please refer to green highlights on page 9-11.
- Method structure might be better using the following structure
- Research model, 2. Data collection 3. Measurement items 4. Analysis
Answer:
The structure has been updated as ordered. Please refer to the method section starting page 11.
Hypotheses number and direction needs to be presented in the research model. If possible, median split analysis for moderating effect is recommended.
Answer:
The research model has been updated as ordered (see Figure 2 page 12). The median split analysis is not facilitated by AMOS (AMOS provides critical ratio differences instead). With 3 groups (dark-green, semi/light-green, and non-green), drawing all pairs in one diagram is quite complicated. Hence, we would like to omit median split analysis in our data presentation.
- Figure presenting results need to be presented separately. Answer:
The Figure presenting result has been separately presented. Please refer to page 24 and 25.
- Theoretical contribution needs to be upgraded as more convincing manners.
Answer:
The Theoretical contribution has been updated. Please refer to the highlights on page 27.
Round 2
Reviewer 1 Report
Comments and Suggestions for Authors
The authors have addressed all the comments. I am fine with the response. and the paper can be accepted.
Comments on the Quality of English Language
Minor editing of English language required